# COMPACT: COMMON-TOKEN OPTIMIZED MODEL PRUNING ACROSS CHANNELS AND TOKENS

## ABSTRACT

Making large language models (LLMs) more efficient in memory, latency, and serving cost is crucial for edge deployment, interactive applications, and sustainable inference at scale. Pruning is a promising technique, but existing pruning methods are limited: width pruning often breaks the standard transformer layout, requiring custom inference code, while depth pruning can cause abrupt accuracy drops. Also, while many pruning approaches are effective against LLMs, they struggle to maintain performance on small language models (SLMs). In this work, we propose COMPACT, which jointly (i) prunes rare vocabulary to shrink embedding/LM head layers and (ii) prunes FFN intermediate channels using *common-token–weighted* activations, aligning importance with the post-pruning token distribution. COMPACT inherits strengths of both depth and width pruning, such as: deployment-friendliness (keeps a standard transformer architecture), scale-adaptivity (trade off vocab. vs. FFN pruning), competitive pruning times, and strong memory savings alongside throughput gains. Experiments across Qwen, LLaMA, and Gemma families (0.5B–70B) show state-of-the-art downstream performance, with substantial reductions in parameters, GPU memory, and latency[1].

## 1 INTRODUCTION

Large language models (LLMs) have achieved remarkable performance across a wide range of natural language tasks, but their ever-growing parameter counts, reaching billions to hundreds of billions, make deployment expensive in terms of memory, inference time, and energy cost. To broaden access and enable real-world applications such as on-device inference, classroom use, or latency-sensitive systems, it is crucial to compress LLMs while retaining as much performance as possible.

Quantization (Frantar et al., 2022; Lin et al., 2024) and pruning (Frantar & Alistarh, 2023; Sun et al., 2024) have been a major line of compression work. This work focuses on structured pruning, removing entire rows and columns of weight matrices. Structured pruning is mainly categorized into depth pruning and width pruning. Depth pruning removes entire transformer blocks (Kim et al., 2024; Song et al., 2024; Gromov et al., 2025), but the coarse-grained removal of layers leads to sharp performance drops. Width pruning trims hidden dimensions such as FFN channels or attention heads (Ma et al., 2023; Ashkboos et al., 2024; An et al., 2024), but they typically deviate from a standard transformer architecture and require custom inference code. In addition, these approaches are limited in three other ways: (i) They prune largely *mechanistically*, without analyzing *where parameters are concentrated* within LLMs (embeddings, FFNs, or attention). This blind pruning means that methods that work for large LLMs often fail for SLMs, as they have different parameter distributions. (ii) They ignore the linguistic nature of NLP models: not all tokens are equally important, yet pruning typically treats all tokens as if they contribute equally. (iii) They often require custom implementation changes to accommodate every model family, making implementation maintenance tedious. These oversights lead to non-robust pruning performance across scales.

To address these issues, we propose COMPACT, a simple but powerful pruning framework with two modules: (i) *Vocabulary pruning* removes rare tokens and shrinks embedding/LM head matrices, directly reducing parameters and memory usage, especially in SLMs. (ii) *Common-token–weighted FFN pruning* further reduces redundancy by scoring channels using activations, but weighting only

---

[1]All code will be released, and the method will be packaged as a plug-in tool.

the common tokens that remain valid after vocabulary pruning. Together, these two complementary modules address the limitations of prior work: *pruning is now guided by parameter distribution, respects the linguistic structure of language tasks, remains compatible with existing inference frameworks, and is architecture-agnostic across most model families*.

We systematically analyze parameter distributions across model families and scales. This reveals a clear pattern: embeddings (vocabulary and LM head layers) are important in SLMs, while FFNs dominate in larger models. This explains why prior pruning methods lack robustness across scales—they prune the same way regardless of where redundancy actually lies. A second insight comes from the statistics of natural language: token frequencies follow a Zipfian distribution (Zhemchuzhina et al., 2022), meaning that rare tokens occur extremely infrequently and contribute little to downstream performance. Removing such rare tokens from the vocabulary reduces embedding size without significantly affecting performance, because language tasks are overwhelmingly driven by common tokens. Together, these observations validate the effectiveness of the COMPACT method.

We evaluate COMPACT on diverse LLM families (Qwen 2.5, LLaMA 3.1/3.2, and Gemma 3) and across scales from 0.5B to 70B parameters. We test on seven downstream benchmarks (MMLU (Hendrycks et al., 2021), HellaSwag (Zellers et al., 2019), WinoGrande (Sakaguchi et al., 2020), ARC-C/E (Clark et al., 2018), PIQA (Bisk et al., 2020), GSM8K (Cobbe et al., 2021)) and also measure pruning time, inference throughput, and GPU memory usage. Our experiments highlight three phenomena: (i) *Scale robustness:* COMPACT maintains state-of-the-art performance at high pruning ratios even for SLMs. (ii) *Smooth degradation:* Unlike depth pruning, which shows abrupt performance drops, COMPACT degrades gracefully with higher pruning. (iii) *End-to-end efficiency:* COMPACT yields substantial GPU memory savings and improved throughput.

Our contributions are threefold: i) We provide a systematic analysis of parameter distribution across embeddings, FFNs, and attention, revealing scale-dependent redundancy that prior pruning methods overlook. ii) We propose COMPACT, a novel pruning method which is linguistically grounded, scale-adaptive, and structure-agnostic. iii) We demonstrate state-of-the-art pruning results across LLM families and scales, showing superior retention on downstream tasks together with clear gains in pruning time, inference efficiency, and GPU memory usage.

Table 1: Advantages of COMPACT.

| | Depth pruning | | | Width pruning | | | COMPACT |
|---|---|---|---|---|---|---|---|
| | ShortGPT | LaCo | LLM-Streamline | SliceGPT | 2SSP | FLAP | (ours) |
| Maintains architecture | ✓ | ✓ | ✓ | | | | ✓ |
| Scale-adaptive | | | | | ✓ | | ✓ |
| Inference speedups | ✓ | ✓ | ✓ | ✓ | | ✓ | ✓ |
| Fast pruning | ✓ | ✓ | | ✓ | ✓ | ✓ | ✓ |
| Architecture-agnostic | | | | | | | ✓ |
| Linguistically grounded | | | | | | | ✓ |

## 2 RELATED WORK

**Depth Pruning** removes entire transformer blocks while preserving the standard architecture and compatibility with common inference frameworks (He et al., 2024; Lu et al., 2024). Representative methods include Shortened LLaMA (perplexity-minimizing), SLEB (iterative recalibration), and angular-similarity pruning (Kim et al., 2024; Song et al., 2024; Gromov et al., 2025). LLM-Streamline trains a lightweight network to recover accuracy but requires hours–days and significant GPUs (Chen et al., 2025). We therefore focus on training-free pruning that runs in minutes on a single GPU; COMPACT can optionally be fine-tuned and outperforms training-based baselines. Because depth pruning is coarse-grained and can cause sharp drops, COMPACT instead prunes rows/columns for a finer-grained alternative.

**Width Pruning** removes hidden dimensions or channels in each layer (Xia et al., 2024; Gao et al., 2024b; Guo et al., 2025). Methods include LLM-Pruner/LoRAPrune (gradient-based) (Ma et al., 2023; Zhang et al., 2024), SliceGPT (orthogonal transforms + low-rank) (Ashkboos et al., 2024), FLAP (stability-based) (An et al., 2024), and Bonsai (perturbation modeling) (Dery et al., 2024).

While effective, they often break the standard transformer layout, requiring custom inference code and limiting deployment. COMPACT avoids these issues by preserving architecture, yielding a deployment-friendly width-pruning method that outperforms depth pruning.

**Vocabulary Size/Pruning.** The vocabulary size of a model is the number of tokens that the model can recognize. Modern LLM vocabularies often reach into the hundreds of thousands and typically remain the same across model scales within a family (Tao et al., 2024). Research into vocabulary size scaling (Tao et al., 2024) has shown that the optimal vocabulary size increases with increasing LLM size, contradicting the common practice of keeping vocabulary size constant over a wide range of model sizes. Prior work prunes vocabularies to tailor vocabulary to a target language/domain (Ushio et al., 2023b; Dorkin et al., 2025; Ushio et al., 2023a; Bogoychev et al., 2024; Yang et al., 2022; 2024b). Others prune the drafter's LM head for speculative decoding speedups (Goel et al., 2025), which does not compress the base model used at inference. In contrast, we (i) perform non-domain-specific vocabulary pruning for general-purpose LLMs, and (ii) couple it with common-token–weighted FFN pruning, so channel scores reflect the token distribution after vocab removal. This keeps a standard Transformer layout, is training-free, and proves robust from 0.5B to 70B.

## 3 PROPOSED METHOD: COMPACT

Before designing an effective pruning strategy (Sections 3.2-3.4 ), we first analyze where parameters are concentrated within modern decoder-only transformers (Section 3.1).

### 3.1 ANALYZING PARAMETER DISTRIBUTION IN LLMS: *Vocabulary* VS. *FFN* VS. *Attention*

Mainstream generative LLMs consist of three major groups of parameters: (i) *vocab parameters*, located in the embedding and LM head layers; (ii) *attention parameters*, from the self-attention blocks; and (iii) *FFN parameters*, from the feed-forward blocks.

Formally, the embedding and LM head layers map between the vocabulary space of size $V$ and the hidden dimension $D$, giving

$$N_{\text{vocab}} = 2VD, \tag{1}$$

(or $VD$ if tied embeddings are used). Each FFN block contains three projection matrices of size $D \times I$, where $I$ is the intermediate dimension, yielding

$$N_{\text{FFN}} = 3LDI, \tag{2}$$

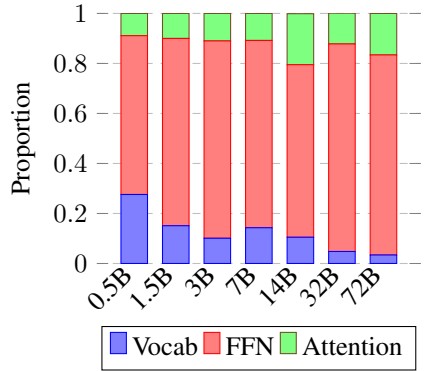

Figure 1: Parameter distribution across Qwen 2.5 models of different scales.

for $L$ layers. For attention, the number of parameters depends on whether grouped query attention is used (Ainslie et al., 2023). With $H$ denoting the ratio of attention heads to KV heads, the count is

$$N_{\text{attention}} = 2LD^2 \left(1 + \frac{1}{H}\right). \tag{3}$$

When $H = 1$, this reduces to $N_{\text{attention}} = 4LD^2$.

Asymptotically, $N_{\text{FFN}}$ and $N_{\text{attention}}$ scale as $O(LD^2)$—$I \approx O(D)$—while $N_{\text{vocab}}$ scales only as $O(D)$, as $V$ is kept constant when scaling. Thus, as model size grows, vocab parameters become proportionally smaller. Conversely, for smaller models, vocab parameters can constitute a significant fraction of the total. We observe this empirically by calculating the relative proportions of each parameter group on popular model families. Figure 1 shows our empirical analysis on the Qwen 2.5 model family (Yang et al., 2024a), which validates our theoretical analysis. Proportions of other model families can be found in Appendix A.1. This motivates our strategy: **vocabulary pruning is an efficient way to reduce parameters, especially in small-to-medium LLMs, while FFN pruning is critical for large models**.

---

**Algorithm 1** COMPACT

---

**Require:** Model $M$, calibration dataset $\mathcal{D}$, target vocabulary size $V'$, target intermediate size $I'$
 1: Identify $S \leftarrow$ set of $V - V'$ rarest tokens in vocabulary.
 2: Run forward passes of $M$ on $\mathcal{D}$, collect squared activations.
 3: For each channel $k$, compute importance $\mathcal{I}_k$ using common act$^2$ (Eq. 5).
 4: **for** each layer **do**
 5:    Prune $I - I'$ least important channels (remove rows of $W_{\text{gate}}, W_{\text{up}}$ and columns of $W_{\text{down}}$).
 6: **end for**
 7: Prune vocab parameters: remove final $V - V'$ rows of embedding and LM head matrices; delete tokenizer merges for tokens in $S$.
 8: **return** pruned model $M'$.

---

### 3.2 FROM RARE TO COMMON: RATIONALE OF VOCABULARY PRUNING

Byte-Pair Encoding (BPE) tokenizers follow Zipf's law (Zhemchuzhina et al., 2022), where most tokens appear extremely rarely. Since BPE builds its vocabulary by merging frequent token pairs, the rarest tokens naturally appear at the end of the vocabulary list.

We define the set $S$ as the $V - V'$ rarest tokens in the vocabulary. These tokens can be directly removed by pruning the corresponding rows in the embedding/LM head matrices and deleting the corresponding merge rules from the tokenizer. The key insight is conceptual: **the deleted tokens will never be generated in the pruned model.** This means that subsequent optimization steps should focus on preserving performance under the *common-token distribution* rather than the full distribution. VOCAB-PRUNING is highly efficient: it requires no calibration data or forward passes.

### 3.3 INTERMEDIATE PRUNING UNDER THE COMMON-TOKEN DISTRIBUTION

Pruning vocabulary parameters alone reduces the embedding size but does not address redundancies in the FFNs, which dominate parameter count in large models. To prune FFNs, we adopt an activation-based criterion. The standard *act$^2$* method (Muralidharan et al., 2024; Sandri et al., 2025) defines the importance of FFN intermediate channel $k$ as

$$\mathcal{I}_k = \sum_{i=1}^{N} \big(\text{SiLU}(X_i W_{\text{gate}}) X_i W_{\text{up}}\big)_k^2, \tag{4}$$

summing squared activations over a calibration dataset. Here, $X_i$ is FFN input and $W_{\text{gate}}, W_{\text{up}}$ are model weights. However, this equally weights all tokens $x_i$, including $x_i \in S$. Since such tokens will never appear in the input after pruning, their activations should not guide channel importance. We therefore introduce *common act$^2$*, a weighted variant:

$$\mathcal{I}_k = \sum_{i=1}^{N} w_i \big(\text{SiLU}(X_i W_{\text{gate}}) X_i W_{\text{up}}\big)_k^2, \quad w_i = \begin{cases} 0 & x_i \in S, \\ 1 & \text{otherwise.} \end{cases} \tag{5}$$

This ensures that FFN pruning is explicitly optimized for the tokens that remain valid after pruning.

### 3.4 COMPACT: JOINT PRUNING PIPELINE

Our proposed method, COMPACT, integrates vocabulary pruning with common act$^2$-based FFN pruning. Importantly, embedding pruning and channel pruning are not performed sequentially in isolation: knowledge of $S$ (rarest tokens) is first identified, then used to guide intermediate pruning, and finally both vocab and FFN parameters are removed. The full pipeline is given in Algorithm 1.

**Advantages of COMPACT.** i) COMPACT *is scale-adaptive.* COMPACT uses two different knobs for pruning: (i) vocabulary pruning at the embedding/LM head layers and (ii) common-token–weighted pruning of FFN intermediate channels. These two knobs are orthogonal, allowing COMPACT to be tunable for SLMs (emphasize vocab pruning) LLMs (emphasize intermediate pruning), or any mix to meet a target budget. This tunability preserves capacity on frequent tokens while enabling strong compression across model scales. ii) COMPACT *is compatible with LLM frameworks.* One weakness of most width pruning methods is that they do not maintain a standard transformer architecture. This is indeed the case with SliceGPT, which prunes the hidden size in all layers except for the

final one, as well as 2SSP, which prunes entire attention modules. As a result, these methods are not compatible with the transformers library, vLLM, or any other inference engines, limiting practicality. In this aspect, COMPACT is similar to depth pruning, since pruning the vocabulary and intermediate size does not affect the transformer architecture. As a result, COMPACT models are compatible with all inference engines, making it a practical approach to width pruning. The full advantages of COMPACT is summarized in Table 1.

## 4 EXPERIMENTS

### 4.1 EXPERIMENTAL SETUP

**Baselines.** We compare with representative and state-of-the-art i) width pruning methods: *SliceGPT*, (Ashkboos et al., 2024), *2SSP* (Sandri et al., 2025); ii) depth pruning methods: *ShortGPT* (Men et al., 2024), *LaCo* (Yang et al., 2024c). All methods use the default calibration set in their paper. For COMPACT, we use 256 calibration samples from the C4 dataset (Raffel et al., 2020).

**LLMs to prune.** We evaluate on a diverse set of LLMs spanning architectures and scales: (i) **SLMs**: *Qwen 2.5–0.5B*, *LLaMA 3.2–1B*, and *Gemma 3–1B*. This mix covers three distinct families, enabling a robustness assessment across architectures; moreover, small LLMs are particularly challenging to prune, as they are often trained beyond the Chinchilla-optimal compute–data balance, leaving limited redundancy. **Nevertheless, pruning small LLMs is highly valuable for edge/on-device use and in privacy- or bandwidth-constrained settings (healthcare, classrooms, federated clients): it shrinks memory/storage, improves end-to-end latency, lowers energy use, and reduces serving cost.** (ii) **LLMs**: *LLaMA 3.1–8B* and *LLaMA 3.1–70B*. Together with the 1B variant above, this suite evaluates pruning effectiveness across a wide scale.

**Evaluation tasks.** Following *SliceGPT* (Ashkboos et al., 2024), we evaluate pruned models using **HellaSwag (HeSw)**, **WinoGrande (WiGr)**, **ARC-C**, **ARC-E**, and **PIQA**. Since Jaiswal et al. (2024) shows that pruned LLMs degrade more on complex tasks, we add **MMLU** for general knowledge, and **GSM8K** for generation tasks. This gives a more complete view of model performance.

**Evaluation Criteria.** Details about the evaluation setup and pruning hyperparameters can be found in Appendix A.2 and Appendix A.3, respectively. (A) We report the percentage of parameters that were removed (**Ratio (%)**), the mean score of the 7 benchmarks (**Avg**), and the relative mean score compared to the dense model (**Avg%**); (B) It is standard to evaluate pruned models on perplexity and downstream tasks. However, because we reduce vocabulary size, our perplexity naturally decreases, making comparisons to baselines unfair. Thus, we only report i) **performance on downstream tasks**, ii) **efficiency regarding pruning time, inference time, and memory usage**.

### 4.2 PERFORMANCE ON DOWNSTREAM TASKS

#### 4.2.1 RESULTS ON SMALLER LLMS

**COMPACT outperforms baselines by large margins.** Our results are summarized in Table 2. Although prior works report strong results on LLMs, they perform poorly on SLMs. On Qwen 2.5–0.5B, GSM8K accuracy collapses for all baselines at only 10% pruning. Likewise, at 10% pruning, MMLU drops to near-random for all models and baselines, with the sole exception of LaCo on Qwen 2.5–0.5B. Because MMLU and GSM8K are the most demanding tasks in our suite, these trends indicate that existing methods fail to preserve performance on challenging benchmarks for SLMs. In contrast, COMPACT delays this collapse: it remains marginally above random on MMLU and GSM8K even at 35% pruning across all models. Across models and pruning ratios, COMPACT attains the highest mean score and leads on nearly all individual benchmarks, despite using a slightly higher pruning ratio than the baselines. Notably, at 35% pruning, COMPACT maintains similar performance to the baselines at 20%, demonstrating superior robustness under high compression.

**COMPACT supports a wide variety of model architectures out-of-the-box.** The official implementations of existing approaches support just a few model architectures. For instance, SliceGPT supports LLaMA/OPT/Phi, LaCo supports LLaMA 2/Baichuan, and ShortGPT only supports LLaMA. Adding support for modern model families like LLaMA 3 and Qwen 2.5 required adding architecture-specific changes, since these architectures can differ significantly in how they handle

Table 2: Pruning SLMs (Qwen 2.5-0.5B, LLaMA 3.2–1B, and Gemma 3–1B) at a ∼10%, ∼20%, and ∼35% ratio. **Please note pruning baselines cannot be applied to Gemma 3**

| | Method | Ratio (%) | MMLU | HeSw | WiGr | ARC-C | ARC-E | PIQA | GSM8K | Avg | Avg% |
|---|---|---|---|---|---|---|---|---|---|---|---|
| **Qwen 2.5-0.5B** | Dense | 0.00 | 47.3 | 52.2 | 56.4 | 32.3 | 58.2 | 69.9 | 34.9 | 50.2 | 100.0 |
| | Random | - | 25.0 | 25.0 | 50.0 | 25.0 | 25.0 | 50.0 | 0.0 | 28.6 | 57.0 |
| | ShortGPT | 9.11 | 27.8 | 44.0 | 53.0 | 25.9 | 45.8 | 66.8 | 0.2 | 37.6 | 75.1 |
| | LaCo | 9.11 | **46.1** | 45.5 | **56.3** | 28.2 | 51.6 | 65.5 | 0.4 | 41.9 | 83.6 |
| | SliceGPT | 10.71 | 23.2 | 43.2 | 53.3 | 26.4 | 50.3 | 63.8 | 0.0 | 37.2 | 74.1 |
| | 2SSP | 10.12 | 25.5 | 46.6 | 54.7 | 27.8 | 52.2 | 68.7 | 1.9 | 39.6 | 79.0 |
| | COMPACT | 11.13 | 45.2 | **51.9** | 55.3 | **32.4** | **59.5** | **70.1** | **28.7** | **49.0** | **97.7** |
| | ShortGPT | 18.02 | 25.0 | 37.7 | 52.0 | 27.1 | 41.7 | 62.1 | 0.0 | 35.1 | 70.0 |
| | LaCo | 18.02 | 24.0 | 36.3 | 49.9 | 23.5 | 41.7 | 62.9 | 0.0 | 34.0 | 67.8 |
| | SliceGPT | 19.64 | 23.1 | 32.1 | 52.0 | 20.2 | 33.4 | 53.7 | 0.0 | 30.6 | 61.1 |
| | 2SSP | 19.64 | 24.3 | 40.9 | 53.8 | 25.3 | 43.9 | 64.3 | 0.5 | 36.1 | 72.0 |
| | COMPACT | 20.24 | **44.1** | **48.1** | **55.4** | **30.6** | **53.3** | **66.6** | **26.3** | **46.3** | **92.4** |
| | ShortGPT | 36.23 | 24.3 | 27.8 | 50.1 | **25.7** | 26.2 | 51.8 | 0.0 | 29.4 | 58.7 |
| | LaCo | 36.23 | 23.9 | 28.2 | 47.9 | 23.9 | 30.6 | 56.2 | 0.0 | 30.1 | 60.0 |
| | SliceGPT | 36.61 | 23.1 | 29.0 | 51.5 | 22.5 | 30.7 | 53.4 | 0.0 | 30.0 | 59.9 |
| | 2SSP | 36.23 | 22.9 | 31.3 | 49.6 | 22.9 | 33.8 | 59.0 | 0.0 | 31.4 | 62.5 |
| | COMPACT | 37.04 | **25.5** | **40.0** | **53.8** | 25.2 | **40.0** | **62.2** | **0.5** | **35.3** | **70.4** |
| **LLaMA 3.2–1B** | Dense | 0.00 | 36.6 | 63.8 | 60.7 | 36.2 | 60.7 | 74.5 | 5.4 | 48.3 | 100.0 |
| | Random | - | 25.0 | 25.0 | 50.0 | 25.0 | 25.0 | 50.0 | 0.0 | 28.6 | 59.2 |
| | ShortGPT | 10.03 | 23.4 | 48.0 | **60.2** | 30.4 | 49.8 | 68.0 | 0.0 | 40.0 | 82.8 |
| | LaCo | 10.03 | 24.4 | 48.4 | 52.0 | 29.5 | 47.9 | 69.3 | 0.5 | 38.9 | 80.5 |
| | SliceGPT | 10.16 | 23.1 | 49.4 | 54.0 | 28.6 | 43.4 | 64.0 | 0.0 | 37.5 | 77.7 |
| | 2SSP | 9.87 | 30.5 | 55.5 | 57.9 | 32.9 | 56.7 | **72.9** | 3.0 | 44.2 | 91.6 |
| | COMPACT | 10.03 | **36.7** | **61.1** | 59.7 | **35.1** | **57.1** | 71.9 | **6.1** | **46.8** | **97.0** |
| | ShortGPT | 19.66 | 22.8 | 40.0 | 55.3 | 29.9 | 35.2 | 58.9 | 0.0 | 34.6 | 71.7 |
| | LaCo | 19.66 | 23.0 | 35.7 | 52.4 | 25.9 | 37.0 | 62.4 | 0.4 | 33.9 | 70.1 |
| | SliceGPT | 20.31 | 23.0 | 39.9 | 52.3 | 26.2 | 38.9 | 58.6 | 0.0 | 34.1 | 70.7 |
| | 2SSP | 19.66 | 26.8 | 46.9 | 53.9 | 27.1 | 50.4 | 68.1 | 2.2 | 39.3 | 81.5 |
| | COMPACT | 19.98 | **30.6** | **54.4** | **58.6** | **32.0** | **51.3** | **69.9** | **3.1** | **42.8** | **88.8** |
| | ShortGPT | 34.47 | 24.3 | 32.5 | 50.0 | **28.4** | 28.9 | 55.4 | 0.0 | 31.4 | 65.0 |
| | LaCo | 34.47 | 23.2 | 37.3 | 50.1 | 23.9 | 29.3 | 55.3 | 0.0 | 29.9 | 61.9 |
| | SliceGPT | 35.16 | 23.0 | 30.4 | 51.2 | 22.0 | 32.9 | 53.4 | 0.0 | 30.4 | 63.1 |
| | 2SSP | 34.63 | 22.9 | 35.1 | 52.6 | 24.5 | 38.9 | **60.9** | 0.0 | 33.6 | 69.6 |
| | COMPACT | 35.03 | **27.9** | **42.8** | **55.6** | 27.7 | **41.7** | 60.6 | **1.8** | **36.9** | **76.4** |
| **Gemma3–1B** | Dense | 0.00 | 24.9 | 62.1 | 59.0 | 38.2 | 71.9 | 74.8 | 2.4 | 47.6 | 100.0 |
| | Random | - | 25.0 | 25.0 | 50.0 | 25.0 | 25.0 | 50.0 | 0.0 | 28.6 | 60.0 |
| | | 10.01 | 24.9 | 60.3 | 59.0 | 39.1 | 69.0 | 74.1 | 1.7 | 46.9 | 98.4 |
| | COMPACT | 20.02 | 25.0 | 55.4 | 59.0 | 37.9 | 63.3 | 70.0 | 1.7 | 44.6 | 93.6 |
| | | 34.99 | 24.2 | 45.1 | 55.9 | 26.5 | 46.4 | 65.3 | 0.5 | 37.7 | 79.2 |

self-attention, layer normalization, etc. The strength of COMPACT is that it only prunes the vocabulary embeddings and FFN blocks, which have been standardized and remain unchanged across the vast majority of model architectures. As a consequence, COMPACT is architecture-agnostic and runs out-of-the-box across many model families. This is most evident with Gemma 3, which uses QK-norm and alternating local and global attention layers—optimizations that prevented us from adapting our baselines. Accordingly, baseline results for Gemma 3 are omitted in Table 2. In contrast, COMPACT operates on Gemma 3 without any architecture-specific changes.

### 4.2.2 RESULTS ON LARGER LLMs

**COMPACT is robust across scales.** Our results are in Table 3. We see that COMPACT achieves state-of-the-art performance for larger models as well, with over 80% performance at a 35% ratio, indicating that our method is highly robust to a wide range of sizes. We attribute this robustness to our dual approach to width pruning. As model size increases, the proportion of vocabulary parameters decreases, which decreases the effectiveness of pruning vocabulary size. However, intermediate pruning becomes more effective as model size increases, similarly to most pruning methods (Xu et al., 2024). These two techniques complement each other's strengths, leading to a robust hybrid pruning method. Although the performance gap between COMPACT and 2SSP narrows at the 70B size, we note that 2SSP's hybrid approach of pruning FFN channels and entire attention blocks deviates from the standard transformer architecture, while COMPACT's approach does not. This makes COMPACT more practical to deploy, while also achieving slightly higher performance.

Table 3: Pruning larger LLMs (LLaMA 3.1–8B & LLaMA 3.1–70B) at a ∼10%, ∼20%, and ∼35% ratio. LaCo failed to prune LLaMA 3.1-70B due to OOM errors, so it is omitted from our results.

|  | Method | Ratio (%) | MMLU | HeSw | WiGr | ARC-C | ARC-E | PIQA | GSM8K | Avg | Avg% |
|---|---|---|---|---|---|---|---|---|---|---|---|
| **LLaMA 3.1–8B** | Dense | 0.00 | 63.4 | 78.9 | 73.6 | 53.4 | 80.9 | 81.1 | 51.6 | 69.0 | 100.0 |
|  | Random | - | 25.0 | 25.0 | 50.0 | 25.0 | 25.0 | 50.0 | 0.0 | 28.6 | 41.4 |
|  | ShortGPT | 10.86 | 58.0 | 73.8 | 70.2 | 47.4 | 71.2 | 77.5 | 29.3 | 61.1 | 88.5 |
|  | LaCo | 10.86 | 58.8 | 73.3 | 72.3 | 48.9 | 73.7 | 76.3 | **32.0** | 62.2 | 90.1 |
|  | SliceGPT | 10.16 | 43.9 | 65.6 | 67.2 | 39.4 | 66.2 | 71.0 | 19.4 | 53.2 | 77.2 |
|  | 2SSP | 10.86 | 54.1 | 74.6 | 71.6 | 46.8 | 71.6 | **79.7** | 14.1 | 58.9 | 85.4 |
|  | COMPACT | 10.00 | **59.6** | **75.2** | **73.7** | **50.3** | **74.9** | 78.4 | 27.8 | **62.9** | **91.1** |
|  | ShortGPT | 19.02 | **58.6** | 64.9 | 68.4 | 42.2 | 58.3 | 71.6 | 0.6 | 52.1 | 75.5 |
|  | LaCo | 19.02 | 24.1 | 54.0 | 55.3 | 29.2 | 51.1 | 72.4 | 0.4 | 40.9 | 59.3 |
|  | SliceGPT | 20.12 | 24.5 | 51.4 | 61.8 | 30.3 | 49.0 | 61.9 | 0.0 | 39.8 | 57.8 |
|  | 2SSP | 19.99 | 37.4 | 67.2 | 68.4 | 38.1 | 61.8 | **76.8** | 4.3 | 50.6 | 73.3 |
|  | COMPACT | 20.00 | 50.7 | **69.9** | **70.1** | **42.8** | **66.0** | 75.9 | **10.8** | **55.2** | **80.0** |
|  | ShortGPT | 35.31 | 23.2 | 34.3 | 59.1 | 29.7 | 36.9 | 57.2 | 0.0 | 34.4 | 49.8 |
|  | LaCo | 35.31 | 23.1 | 34.8 | 53.1 | 27.3 | 31.5 | 58.8 | 0.0 | 32.7 | 47.3 |
|  | SliceGPT | 35.16 | 23.0 | 35.0 | 54.3 | 23.5 | 37.2 | 55.0 | 0.0 | 32.6 | 47.2 |
|  | 2SSP | 34.77 | 25.3 | 49.9 | 59.3 | 27.1 | 44.3 | 68.7 | **2.3** | 39.5 | 57.3 |
|  | COMPACT | 34.99 | **35.9** | **56.0** | **63.3** | **30.8** | **48.4** | **70.6** | 1.7 | **43.8** | **63.5** |
| **LLaMA 3.1–70B** | Dense | 0.00 | 75.2 | 85.0 | 79.5 | 64.7 | 86.7 | 84.4 | 80.6 | 79.4 | 100.0 |
|  | Random | - | 25.0 | 25.0 | 50.0 | 25.0 | 25.0 | 50.0 | 0.0 | 28.6 | 36.0 |
|  | ShortGPT | 9.77 | **75.0** | 82.9 | 78.5 | 60.8 | 84.8 | 83.4 | 74.5 | 77.1 | 97.1 |
|  | SliceGPT | 10.06 | 70.6 | 75.4 | 76.6 | 58.3 | 82.0 | 79.7 | 62.4 | 72.1 | 90.8 |
|  | 2SSP | 9.92 | 73.4 | **84.7** | 78.6 | 63.9 | 85.4 | **84.2** | 75.1 | 77.9 | 98.1 |
|  | COMPACT | 10.06 | 73.5 | 84.5 | **79.5** | **64.0** | **86.0** | 84.1 | **76.0** | **78.2** | **98.5** |
|  | ShortGPT | 20.68 | **74.9** | 79.3 | **78.0** | 56.0 | 80.1 | 80.1 | 53.0 | 71.6 | 90.2 |
|  | SliceGPT | 20.02 | 63.1 | 64.8 | 74.0 | 52.4 | 76.7 | 73.7 | 0.0 | 57.8 | 72.8 |
|  | 2SSP | 20.11 | 68.8 | **83.7** | 76.2 | 59.5 | 82.1 | **83.7** | 62.1 | 73.7 | 92.8 |
|  | COMPACT | 20.11 | 70.6 | 83.3 | 76.2 | **59.7** | **82.6** | **83.7** | **62.6** | **74.1** | **93.3** |
|  | ShortGPT | 36.40 | **71.2** | 66.5 | **75.7** | 46.8 | 69.6 | 72.3 | 0.0 | 57.4 | 72.3 |
|  | SliceGPT | 35.06 | 29.3 | 37.1 | 66.6 | 34.5 | 59.3 | 62.6 | 0.0 | 41.3 | 52.0 |
|  | 2SSP | 35.13 | 58.5 | 77.7 | 72.1 | 50.7 | 74.2 | **81.6** | **25.0** | 62.8 | 79.1 |
|  | COMPACT | 34.99 | 59.6 | **78.1** | 72.8 | **53.2** | **76.1** | 81.3 | **25.0** | **63.7** | **80.2** |

**COMPACT shows smooth degradation.** On LLaMA 3.1–8B, we observe that ShortGPT surpasses COMPACT at a 20% pruning ratio, but not at 10% or 35%, with its 20% score even exceeding its 10% score. This aligns with the *step-like* behavior in Gromov et al. (2025) for depth-pruning methods, where performance remains intact up to a critical threshold and then collapses abruptly. ShortGPT also exhibits this pattern, explaining the spike at 20%. In contrast, COMPACT (a width-pruning method) shows smooth MMLU degradation as pruning increases—hence the dip relative to ShortGPT at 20%, followed by recovery and a lead at 35%. Even at 35% pruning, COMPACT remains above random on both MMLU and GSM8K. A similar effect is seen on LLaMA 3.1–70B.

**Analysis: Why our pruning hurts performance minimally?** Changing the vocabulary affects how text is tokenized: If a token is removed during pruning, the text associated with the token is now tokenized as multiple shorter, more common tokens. We analyze how often rare tokens occur. We use the questions in each of our benchmarks, as well as 10k random samples from the C4 dataset. Our results are in Table 4. Our pruned model reduces vocabulary size from 150k to 50k, a 67% reduction. Despite this, only 4% of words are tokenized differently from the original, regardless of text source. These results provide insight to the effectiveness of VOCAB-PRUNING: significant proportions of vocabulary only affects a small fraction of text, so removing these vocabulary has little impact on performance.

Table 4: Proportion of words re-tokenized after 35% pruning of Qwen 2.5–0.5B, by dataset.

| Dataset | Rare% |
|---|---|
| **MMLU** | 4.43% |
| **HellaSwag** | 3.48% |
| **WinoGrande** | 5.01% |
| **ARC-C** | 3.82% |
| **ARC-E** | 3.95% |
| **PIQA** | 5.68% |
| **GSM8K** | 3.60% |
| **C4** | 4.56% |

### 4.3 EFFICIENCY IN PRUNING TIME, INFERENCE LATENCY, AND MEMORY

**Pruning time.** The main strength of training-free methods is that they can prune large models efficiently. To test this, we report pruning times at 35% pruning on LLaMA 3.1-8B and 70B to best discriminate between methods.

Table 6: Throughput and memory usage for LLaMA 3.1-8B.

| | Method | Ratio | Memory Usage (MB) | Throughput (q/s) |
|---|---|---|---|---|
| | Dense | 0.00% | 50030 | 147.01 |
| Classification | ShortGPT/LaCo | 35.31% | 44624 (0.89x) | **221.61 (1.51x)** |
| | 2SSP | 34.77% | 44985 (0.90x) | 104.03 (0.71x) |
| | SliceGPT | 35.16% | 42440 (0.85x) | 173.94 (1.18x) |
| | COMPACT | 34.99% | **32066 (0.64x)** | 201.19 (1.37x) |
| | Dense | 0.00% | 21787 | 81.18 |
| Generation | ShortGPT/LaCo | 35.31% | 16336 (0.75x) | **128.03 (1.57x)** |
| | COMPACT | 34.99% | **14248 (0.65x)** | 112.40 (1.38x) |

Table 7: Comparison of post-pruning recovery fine-tuning methods.

| Method | Params (M) | MMLU | HeSw | WiGr | ARC-C | ARC-E | PIQA | GSM8K | Avg | Avg% |
|---|---|---|---|---|---|---|---|---|---|---|
| Dense | 494 | 47.3 | 52.2 | 56.4 | 32.3 | 58.2 | 69.9 | 34.9 | 50.2 | 100.0 |
| Random | - | 25.0 | 25.0 | 50.0 | 25.0 | 25.0 | 50.0 | 0.0 | 28.6 | 57.0 |
| LLM-Streamline | 315 | 23.0 | 31.8 | 53.4 | 23.5 | 37.9 | 59.7 | 0.2 | 32.8 | 65.4 |
| Gemma 3-270M | 268 | 26.2 | 41.5 | 53.8 | 28.4 | **57.2** | **68.4** | 1.2 | 39.5 | - |
| Our COMPACT | 311 | 25.5 | 40.0 | 53.8 | 25.2 | 40.0 | 62.2 | 0.5 | 35.3 | 70.4 |
| + CPT | 311 | 25.8 | **45.1** | **55.8** | 27.9 | 52.7 | 67.7 | 0.0 | 39.3 | 78.3 |
| + SDD | 311 | **39.7** | 43.0 | 54.9 | **28.6** | 50.6 | 65.6 | **11.3** | **41.9** | **83.6** |

Our results are in Table 5. With LLaMA 3.1-8B, COMPACT is 3 times faster than 2SSP, our strongest baseline, and comparable our depth pruning methods, with pruning times under a minute. At the 70B size, COMPACT now becomes 6 times faster than 2SSP. The low pruning times show that COMPACT has competitive efficiency to our baselines.

**Inference speed and memory usage.** We evaluate two inference paradigms: *Text classification* and *Text generation*. The inference setup is in Appendix A.4.

Our results are in Table 6. Note that ShortGPT and LaCo prune the same number of layers, so they have the same inference performance. In the text classification task, our method achieves the highest memory reduction, and the second-highest throughput increase. The low memory usage is from

Table 5: 3-run average pruning time (mm:ss) comparison at a 35% pruning ratio for LLaMA 3.1. We exclude I/O time for fairness.

| | Method | Pru. Time |
|---|---|---|
| | ShortGPT | 0:18 |
| | LaCo | 0:05 |
| 8B | SliceGPT | 10:48 |
| | 2SSP | 1:26 |
| | COMPACT | 0:32 |
| | ShortGPT | 2:10 |
| | SliceGPT | 84:38 |
| 70B | 2SSP | 13:48 |
| | COMPACT | 2:17 |

the reduced vocabulary size. During the forward pass, logits are stored in GPU memory, which becomes very large with high batch sizes. By pruning the vocabulary size, COMPACT shrinks logit size, causing large memory reductions. **This is especially important in edge computing applications, where memory is very limited and can spell the difference in whether a model can be used or not.** COMPACT has higher throughput than our width pruning baselines SliceGPT/2SSP, although it falls behind depth pruning methods. This aligns with Bian et al. (2025), which showed that scaling down layer count proportionally increases throughput, but scaling layer size does not. While COMPACT achieves faster inference than other width pruning methods, more work is needed to match the throughput of depth pruning methods. In the generation task, the trends in memory usage and throughput are similar to that of the classification task.

## 4.4 RECOVERY FINE-TUNING

Although COMPACT is training-free, we optionally apply recovery fine-tuning. We fine-tune Qwen 2.5–0.5B pruned at 35% using two approaches: (i) Continued Pretraining (CPT): train on 900M tokens from FineWeb-Edu (Penedo et al., 2024); (ii) Self-Data Distillation (SDD): generate 900M tokens from the unpruned model (temperature = 1.0) to match its training distribution, then fine-tune on this synthetic data (Thangarasa et al., 2024).

Despite some incoherence in the SDD synthetic dataset, our results (Table 7) show that SDD consistently improves all benchmarks and yields a higher average score than CPT, which fails to boost MMLU or GSM8K. This confirms SDD's effectiveness even below 1B parameters (Thangarasa et al., 2024). With fine-tuning, COMPACT surpasses training-based LLM-Streamline (Chen et al., 2025) and even outperforms Gemma 3-270M, pretrained on 6T tokens—highlighting pruning's efficiency relative to pretraining.

Table 8: Multiple values of $V'$ and $I'$ over a fixed pruning ratio for Qwen 2.5-0.5B.

|  | V' | I' | Ratio | Avg | Avg% |
|---|---|---|---|---|---|
| Dense | 151936 | 4864 | 0.00% | 50.2 | 100.0 |
| ACT$^2$ | 151936 | 2048 | 36.84% | 31.6 | 63.0 |
|  | 131584 | 2304 | 37.04% | 31.8 | 63.3 |
|  | 111104 | 2560 | 37.45% | 33.1 | 66.1 |
|  | 90752 | 2944 | 36.23% | 34.2 | 68.2 |
|  | 70400 | 3200 | 36.44% | 34.5 | 68.9 |
| COMPACT | 49536 | 3456 | 37.04% | **35.3** | **70.4** |
|  | 29568 | 3712 | 37.25% | 34.7 | 69.2 |
|  | 9088 | 3968 | 37.65% | 32.4 | 64.7 |

## 5 ABLATION STUDY

In our ablation study, we try to further four questions: i) $\mathcal{Q}_1$: how effective is COMMON-ACT$^2$? ii) $\mathcal{Q}_2$: how to trade off VOCAB-PRUNING and FFN-PRUNING? iii) $\mathcal{Q}_3$: how much calibration data?

**Answer to $\mathcal{Q}_1$: effectiveness of COMMON-ACT$^2$.**
We compare our novel COMMON-ACT$^2$ method with ACT$^2$. To isolate the effect of the intermediate pruning method, we prune rare vocabulary for all models, then apply the specified intermediate pruning method. We also test another commonly used method, |act|, which is similar to ACT$^2$ but uses the

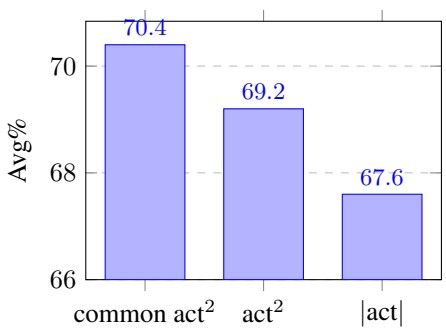

Figure 2: COMMON-ACT$^2$ outperforms both ACT$^2$ and |act| (Qwen 2.5-0.5B at a 35% pruning ratio).

summed *absolute* activations instead of squared activations. Our results are summarized in Figure 2. We find that COMMON-ACT$^2$ achieves the highest mean performance compared to our baselines.

**Answer to $\mathcal{Q}_2$: vocabulary-intermediate tradeoff** COMPACT has two hyperparameters: the new vocabulary size $V'$ and the new intermediate size $I'$. These can be adjusted to produce many configurations at a given pruning ratio. We test different configurations of Qwen 2.5-0.5B at 35% pruning. Our results are in Table 8. Note that $V' = 151936, I' = 2048$ is identical to ACT$^2$. Despite ACT$^2$'s simplicity, it achieves better performance than our baselines even without VOCAB-PRUNING. However, the best result is achieved with a combination of both, validating COMPACT's methodology.

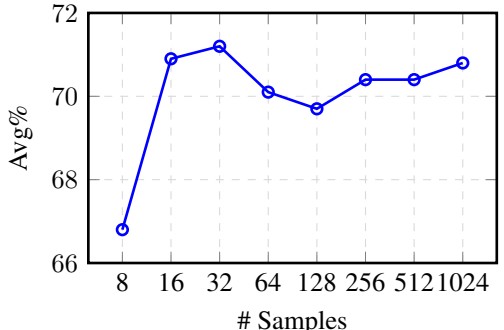

Figure 3: Downstream performance by calibration dataset size. Although all benchmarks used 256 calibration samples, 16 samples is sufficient.

**Answer to $\mathcal{Q}_3$: calibration data size.** We perform ablations over the number of calibration samples, with our results in Figure 3. COMPACT is highly robust to sample count, and similar performance can be achieved with just 16 samples, implying that the pruning time can be reduced further.

## 6 CONCLUSION

We propose COMPACT, a training-free pruning method combining vocabulary and FFN pruning under the common-token distribution. Experiments show that it achieves robust performance across model scales, offering smooth degradation, strong efficiency, and broad deployment compatibility. In future works, we plan to address the discrepancy in throughput between our method and Short-GPT/LaCo, closing the gap between width and depth pruning.

**LLM Usage Disclosure.** We used GPT-5 to assist with language polishing of the manuscript. No parts of the methodology, experimental design, or results were generated by an LLM.

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

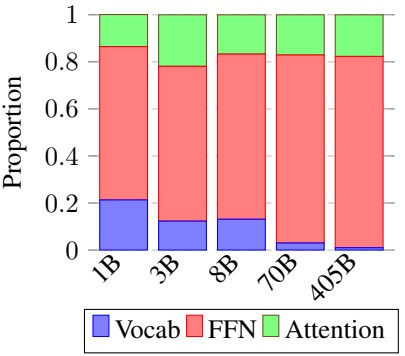

Figure 4: Llama 3 Parameter Distribution

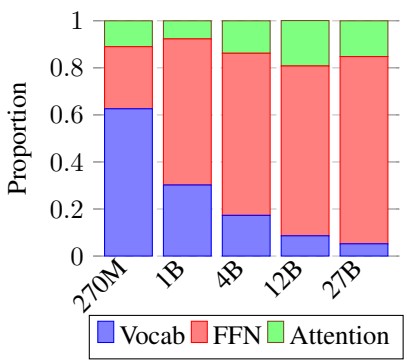

Figure 5: Gemma 3 Parameter Distribution

Yifei Yang, Zouying Cao, and Hai Zhao. Laco: Large language model pruning via layer collapse. In Yaser Al-Onaizan, Mohit Bansal, and Yun-Nung Chen (eds.), *Findings of the Association for Computational Linguistics: EMNLP 2024, Miami, Florida, USA, November 12-16, 2024*, pp. 6401–6417. Association for Computational Linguistics, 2024c. doi: 10.18653/V1/2024.FINDINGS-EMNLP.372. URL https://doi.org/10.18653/v1/2024.findings-emnlp.372.

Ziqing Yang, Yiming Cui, and Zhigang Chen. Textpruner: A model pruning toolkit for pre-trained language models. In Valerio Basile, Zornitsa Kozareva, and Sanja Stajner (eds.), *Proceedings of the 60th Annual Meeting of the Association for Computational Linguistics, ACL 2022 - System Demonstrations, Dublin, Ireland, May 22-27, 2022*, pp. 35–43. Association for Computational Linguistics, 2022. doi: 10.18653/V1/2022.ACL-DEMO.4. URL https://doi.org/10.18653/v1/2022.acl-demo.4.

Rowan Zellers, Ari Holtzman, Yonatan Bisk, Ali Farhadi, and Yejin Choi. Hellaswag: Can a machine really finish your sentence? In Anna Korhonen, David R. Traum, and Lluís Màrquez (eds.), *Proceedings of the 57th Conference of the Association for Computational Linguistics, ACL 2019, Florence, Italy, July 28- August 2, 2019, Volume 1: Long Papers*, pp. 4791–4800. Association for Computational Linguistics, 2019. doi: 10.18653/V1/P19-1472. URL https://doi.org/10.18653/v1/p19-1472.

Mingyang Zhang, Hao Chen, Chunhua Shen, Zhen Yang, Linlin Ou, Xinyi Yu, and Bohan Zhuang. Loraprune: Structured pruning meets low-rank parameter-efficient fine-tuning. In Lun-Wei Ku, Andre Martins, and Vivek Srikumar (eds.), *Findings of the Association for Computational Linguistics, ACL 2024, Bangkok, Thailand and virtual meeting, August 11-16, 2024*, pp. 3013–3026. Association for Computational Linguistics, 2024. doi: 10.18653/V1/2024.FINDINGS-ACL.178. URL https://doi.org/10.18653/v1/2024.findings-acl.178.

Elizaveta Zhemchuzhina, Nikolai Filippov, and Ivan P. Yamshchikov. Pragmatic constraint on distributional semantics. *CoRR*, abs/2211.11041, 2022. doi: 10.48550/ARXIV.2211.11041. URL https://doi.org/10.48550/arXiv.2211.11041.

# A APPENDIX

## A.1 PARAMETER DISTRIBUTION ACROSS OTHER MODEL FAMILIES

Figures 4 and 5 provide parameter distributions for the LLaMA 3 and Gemma 3 model families, respectively. We see that these models follow the same trend as Qwen 2.5 where SLMs have a higher proportion of vocabulary parameters, corroborating our theoretical analysis.

## A.2 EVALUATION METHODOLOGY

Downstream evaluations were conducted using the LM-Evaluation-Harness (Gao et al., 2024a), specifically lm-eval 0.4.8. The evaluation details for each benchmark are in Table 9.

Table 9: Evaluation methodology for each benchmark.

| Benchmark | n-shot | Type | Metric |
|---|---|---|---|
| MMLU | 0 | multiple-choice | acc |
| HellaSwag | 0 | multiple-choice | acc_norm |
| WinoGrande | 0 | multiple-choice | acc |
| ARC-C | 0 | multiple-choice | acc_norm |
| ARC-E | 0 | multiple-choice | acc_norm |
| PIQA | 0 | multiple-choice | acc_norm |
| GSM8K | 5 | generative | strict_match |

Table 10: COMPACT pruning hyperparameters for all main results.

| | Ratio (%) | V' | I' |
|---|---|---|---|
| **Qwen 2.5–0.5B** | 0.00 | 151936 | 4864 |
| | 10.00 | 99968 | 4736 |
| | 20.00 | 49536 | 4736 |
| | 35.00 | 49536 | 3456 |
| **LLaMA 3.2–1B** | 0.00 | 128256 | 8192 |
| | 10.00 | 67968 | 8192 |
| | 20.00 | 56704 | 7168 |
| | 35.00 | 33792 | 5760 |
| **Gemma 3–1B** | 0.00 | 262144 | 6912 |
| | 10.00 | 174592 | 6912 |
| | 20.00 | 86912 | 6912 |
| | 35.00 | 95232 | 5120 |
| **LLaMA 3.1–8B** | 0.00 | 128256 | 14336 |
| | 10.00 | 73216 | 13440 |
| | 20.00 | 67328 | 11520 |
| | 35.00 | 67840 | 8448 |
| **LLaMA 3.1–70B** | 0.00 | 128256 | 28672 |
| | 10.00 | 112384 | 25216 |
| | 20.00 | 111872 | 21632 |
| | 35.00 | 110976 | 16256 |

### A.3 PRUNING HYPERPARAMETERS

We provide $V'$ and $I'$ for all our main results in Table 10. These hyperparameters were found by sweeping over all possible configurations for the given pruning ratio, similarly to Table 8.

### A.4 INFERENCE SETTINGS

Settings: i) **Text classification:** When running our pruned models on our downstream performance benchmarks, we record the maximum memory usage as well as the throughput, measured in number of questions per second. For this test, we use the HellaSwag benchmark, as it is the longest test in our benchmark suite, which allows us to better discriminate between the methods. We test on a larger model (LLaMA 3.1-8B) using a 3-run average, again to better discriminate between methods. All models are loaded in 16-bit precision on a single A100-80GB GPU, with a batch size of 256. ii) **Text generation:** We use the vLLM library (Kwon et al., 2023) to test the memory reduction and inference speedup of our method. As mentioned before, because SliceGPT and 2SSP are incompatible with vLLM, we omit it from our tests. Similarly to the text classification test, we use a 3-run average of LLaMA 3.1-8B in 16-bit precision on a single A100-80GB GPU, but with a batch size of 1 instead, as this is a more realistic workload for on-device text generation. Tests are conducted with 128 input tokens and 128 output tokens.

