# OpenReview forum: "COMPACT: Common-token Optimized Model Pruning Across Channels and Tokens"
_ICLR.cc/2026/Conference — ICLR 2026 Conference Withdrawn Submission_

### Official Review · Reviewer_vgbx · 2025-10-25

**Soundness:** 3
**Presentation:** 3
**Contribution:** 2
**Rating:** 4
**Confidence:** 5

**Summary:**

This paper introduces COMPACT, a training-free pruning framework for large language models to enhance their efficiency in terms of memory, latency, and computational cost. The core innovation of COMPACT lies in two pruning techniques: one is the vocabulary pruning, and the other is the common-token-weighted FFN pruning. Experiments on a range of LLM families (Qwen, LLaMA, and Gemma) demonstrate that COMPACT reduces the parameter count while maintaining strong performance on downstream tasks.

**Strengths:**

1. The paper proposes a training-free and efficient pruning method. The paper reports competitive pruning times, making it a practical solution when dealing with large-scale models.
2. The paper presents strong empirical results. The experimental evaluation is comprehensive, covering multiple model families and a wide range of model sizes.
3. The paper exhibits robustness on small language models. Compact demonstrates superior performance on the smaller models, which is important on edge and resource-constrained devices.

**Weaknesses:**

1. The effectiveness of vocabulary pruning is questionable. The token embedding accounts for only a small part of the entire computation, and shrinking the vocabulary size does not reduce the FLOPs. Therefore, vocabulary pruning only improves the memory usage, while not improving the inference latency. In fact, most model pruning methods focus on removing non-embedding parameters, which speeds up the model more significantly.
2. The pruning method is designed according to the proportion of vocabulary, attention, and FFN parameters, whereas the impact factor of the parameters is a more important assessment. What if the parameters of attention have little impact on the final predictions? The design of the pruning method requires a more solid motivation.
3. The paper claims that COMPACT inherits the strengths of depth pruning, while the method does not involve any depth compression. This is a bit confusing. Although in Line 218, the author claims that COMPACT is similar to depth pruning in keeping the consistent transformer architecture, it is not the case, as channel pruning can keep the transformer architecture as well [1].
4. While the combination and weighting strategy are novel, the individual components of vocabulary pruning and activation-based FFN pruning are existing techniques. The primary contribution lies in the effective and synergistic integration of these methods.
5. The paper lacks comparison with the latest advances in structured LLM pruning [1][2][3].


[1] Lin C H, Gao S, Smith J S, et al. MoDeGPT: Modular Decomposition for Large Language Model Compression[C]//The Thirteenth International Conference on Learning Representations, 2025.
[2] Ling G, Wang Z, Liu Q. Slimgpt: Layer-wise structured pruning for large language models[J]. Advances in Neural Information Processing Systems, 2024, 37: 107112-107137.
[3] Wang X, Zheng Y, Wan Z, et al. SVD-LLM: Truncation-aware Singular Value Decomposition for Large Language Model Compression[C]//The Thirteenth International Conference on Learning Representations, 2025.

**Questions:**

1. How to decide the pruning hyperparameters on different models? While the paper provides the hyperparameters used in their experiments, the process of finding these optimal values for new models could be complex.
2. The author(s) claim that they report the inference latency, while in fact they report the throughput of the model. This inaccuracy diminishes the academic rigor and professionalism of the paper.

---

### Official Review · Reviewer_jQpV · 2025-10-28

**Soundness:** 2
**Presentation:** 2
**Contribution:** 2
**Rating:** 2
**Confidence:** 5

**Summary:**

This paper introduces a training-free pruning framework (COMPACT) aimed at improving the efficiency of LLMs. COMPACT jointly prunes rare tokens from the vocabulary to shrink embedding and LM head layers, and prunes intermediate FFN channels using activation scores weighted toward common tokens that remain after vocabulary pruning. The method is designed to preserve the standard transformer architecture, making it compatible with existing inference frameworks. Evaluations across multiple model families and scales from 0.5B to 70B parameters show that it achieves smooth performance degradation, and consistent results across model sizes.

**Strengths:**

- COMPACT introduces a conceptually simple yet general pruning framework that jointly addresses vocabulary and FFN redundancy while preserving the standard transformer architecture, ensuring compatibility with existing inference frameworks.

- The paper proposes a linguistically grounded perspective on pruning, which acts as an interesting new regime.

- It achieves smooth performance degradation under increasing pruning ratios.

**Weaknesses:**

- The motivation behind vocabulary pruning is not fully convincing. From Figure 1, it appears that the FFN layers dominate the parameter distribution, across scales. Given current trends toward deploying models above 3B parameters, the practical benefit of pruning vocabulary—whose contribution to total parameters rapidly diminishes—seems limited. The paper could better justify why vocabulary pruning remains a relevant or high-impact approach in modern LLM settings.

- The authors claim that COMPACT scales from 0.5B to 70B models, but the main benefits (vocabulary pruning) are most significant for smaller models. For large models, FFN pruning dominates, making the proposed “common-token” weighting a marginal modification over existing approaches. The contribution thus feels incremental.

- Despite claims of “throughput gains,” the reported results show no clear or consistent improvements compared to other baselines. In fact, throughput benefits mainly arise from vocabulary size reduction rather than architectural pruning. The paper should clarify how COMPACT achieves speedups independent of vocabulary compression, especially for models ≥3B where embedding layers are a small portion of runtime cost.

- The central premise of this paper revolves around vocabulary size reduction in small language models (SLMs); however, the reported evaluations of throughput and memory gains are conducted primarily on LLMs. Given the paper’s own claim that vocabulary parameters constitute a larger fraction of total parameters in SLMs, one would expect COMPACT to demonstrate more pronounced memory savings and potentially greater throughput improvements specifically in that regime.

- Despite this work focusing on vocabulary pruning, all benchmarks are English-language tasks. No multilingual or long-context evaluations are included, which weakens claims of linguistic generality.

- The comparison among pruning baselines may be unfair because different calibration methods or datasets are used. Since pruning criteria such as activation magnitude are sensitive to calibration data, a controlled experiment—where all methods share identical calibration samples—is necessary to validate fairness. This control is currently missing or insufficiently discussed.

- Vocabulary pruning may fundamentally limit linguistic expressivity—one of the core strengths of LLMs. By removing rare tokens, the model risks producing more redundant or generic outputs and may underperform on future tasks or even reasoning performance that relies on nuanced or less frequent vocabulary to articulate complex ideas.

**Questions:**

Language and Grammatical Errors:
- The paper is riddled with several grammatical errors (particularly in the introduction) and the authors must strive to correct them to improve readability.

- Incorrect use of LLMs and SLMs: I see several sentences mentioning terms like small LLMs, which expands to: "small large language models". Please note that L in LLMs is for large and S in SLMs is small.

- The variable V' is introduced abruptly in Algorithm 1 without prior explanation or definition in the main text, which is poor practice and hampers readability.

Technical concerns and points to address in rebuttal:
- Related work is lacking, and the authors should include more works such as [1],[2] and [3] in their literature survey.

- Line 45: "They ignore the linguistic nature ....": Is this actually true ? When WANDA [1] applies its pruning metric it is including activations as well, so how are all tokens treated equally ? The importance of weight and activation combined together determines the pruning metric. And similarly with [3].

- Conduct experiments for throughput and memory for SLMs.

- I implore the authors to conduct experiments on reasoning SLMs and LLMs to see how this vocabulary pruning truly generalizes.

- Sweeping through different values of V' (vocabulary size) and I' (intermediate dimension) appears time-intensive, particularly for newer model families or models with evolving tokenizers and embedding layers. This search overhead should arguably be included in the reported pruning time to provide a realistic measure of end-to-end efficiency. Moreover, the generality of COMPACT is uncertain when applied to models with substantially different tokenization schemes or embedding architectures.

- Experiments on multi-lingual task is a requirement for this method.

- Experiment with uniform calibration data choice and/or ablation to show why using different calibration choices does not show differing performance.

References:

[1] Sun, Mingjie, et al. "A simple and effective pruning approach for large language models." arXiv preprint arXiv:2306.11695 (2023).

[2] Yin, L., Wu, Y., Zhang, Z., Hsieh, C. Y., Wang, Y., Jia, Y., ... & Liu, S. (2023). Outlier weighed layerwise sparsity (owl): A missing secret sauce for pruning llms to high sparsity. arXiv preprint arXiv:2310.05175.

[3] Ramachandran, A., Kundu, S., Raha, A., Kundu, S., Mathaikutty, D. K., & Krishna, T. (2025). Accelerating llm inference with flexible n: M sparsity via a fully digital compute-in-memory accelerator. arXiv preprint arXiv:2504.14365.

---

### Official Review · Reviewer_bogr · 2025-10-30

**Soundness:** 3
**Presentation:** 2
**Contribution:** 3
**Rating:** 4
**Confidence:** 3

**Summary:**

This paper studies pruning method for Large Language Models. The proposed method combines vocabulary pruning with Feed-Forward Network (FFN) pruning. It also studies the parameter distribution across models with different scales. Extensive experiments are conducted, and show the validity of the proposed method.

**Strengths:**

1. Showing the parameter distribution across different model scales in beneficial to the community.
2. Extensive experiments are conducted, e.g., using multiple models with different scales.
3. The proposed method is effective and shows many advantages like deployment friendliness.

**Weaknesses:**

1. The presentation should be improved. Current presentation is not clearly enough, making some details hard to follow. E.g., the SiLU in Eq. 4 and 5 is not defined. There are too many bold sentences in the manuscript.
2. The proposed method is adhoc and should be presented in a more principled way to emphasize the technical contribution.
3. The proposed method should provide a method for allocating the pruning budget between vocabulary and FFNs for a given model size and target ratio.

**Questions:**

1. the authors are suggested to carefully polish the presentation.
2. More discussions are required on the tradeoff of pruning between vocabulary and FFNs.

---

### Official Review · Reviewer_Z5gt · 2025-11-02

**Soundness:** 3
**Presentation:** 3
**Contribution:** 3
**Rating:** 4
**Confidence:** 4

**Summary:**

The paper points out the inability of existing LLM pruning methods to balance performance retention, deployment compatibility, and scale adaptivity across small and large models. The paper propose COMPACT, a training-free pruning framework that combines two complementary modules: (1) vocabulary pruning by removing rare tokens to shrink embedding/LM head layers, optimized for SLMs and (2) common-token weighted FFN pruning using a modified common-act² criterion to prune redundant FFN channels, tailored for LLMs.

The method is grounded in two key insights: 1) parameter distribution is scale-dependent and 2) natural language tokens follow a Zipfian distribution where rare tokens contribute minimally to performance. Experiments across Qwen 2.5, LLaMA 3.1/3.2, Gemma 3  and 7 downstream benchmarks show COMPACT outperforms depth/width pruning baselines in performance retention, while maintaining standard Transformer architecture and competitive efficiency.

**Strengths:**

1. The evaluation covers diverse models, scales, and benchmarks

2. COMPACT preserves the original layout of standard Transformer architectures, which makes it compatible with off-the-shelf frameworks.

3. COMPACT is training-free and requires minimal calibration data (16 samples suffice), and supports recovery fine-tuning to further boost performance.

**Weaknesses:**

1. A motivation is edge deployment, but efficiency experiments (memory, throughput) are conducted on an A100 GPU not edge hardware.

2. The recovery fine-tuning section focuses on SDD and CPT but does not compare to parameter-efficient methods (e.g., LoRA, QLoRA) that are widely used for pruning recovery.

**Questions:**

1. When identifying rare tokens for vocabulary pruning, the paper mentions using token frequency. Does this frequency refer to statistics from the model’s original pre-training dataset, or a specific auxiliary dataset (e.g., C4) used in the experiments?

2. For the common tokens in the common-act² criterion, how is the threshold for defining common determined ?

3. When pruning FFN channels, does COMPACT apply the same pruning ratio (I-I') to all FFN layers in a model, or does it use a layer-wise adaptive ratio ?

---

### Note · Authors · 2026-01-12

I have read and agree with the venue's withdrawal policy on behalf of myself and my co-authors.